# NaCl Ionization-Based Moisture Sensor Prepared by Aerosol Deposition for Monitoring Respiratory Patterns

**DOI:** 10.3390/s22145178

**Published:** 2022-07-11

**Authors:** Myung-Yeon Cho, Ik-Soo Kim, Min-Ji Kim, Da-Eun Hyun, Sang-Mo Koo, Hiesang Sohn, Nam-Young Kim, Sunghoon Kim, Seunghoon Ko, Jong-Min Oh

**Affiliations:** 1Department of Electronic Materials Engineering, Kwangwoon University, 20 Kwangwoon-ro, Nowon-gu, Seoul 01897, Korea; cmy7540@kw.ac.kr (M.-Y.C.); vladimir1219@naver.com (M.-J.K.); hde1704@ktl.re.kr (D.-E.H.); smkoo@kw.ac.kr (S.-M.K.); 2Department of Materials Science and Engineering, Pohang University of Science and Technology, 77 Cheongam-ro, Pohang 37673, Korea; kimiksoo@postech.ac.kr; 3Department of Chemical Engineering, Kwangwoon University, 20 Kwangwoon-ro, Nowon-gu, Seoul 01897, Korea; hsohn@kw.ac.kr; 4RFIC Center, Kwangwoon University, 20 Kwangwoon-ro, Nowon-gu, Seoul 01897, Korea; nykim@kw.ac.kr; 5Department of Applied Chemistry, Dong-Eui University, Busan 47227, Korea; hoon@deu.ac.kr

**Keywords:** respiration monitoring, moisture sensor, NaCl/BaTiO_3_, aerosol deposition, ionization

## Abstract

A highly polarizable moisture sensor with multimodal sensing capabilities has great advantages for healthcare applications such as human respiration monitoring. We introduce an ionically polarizable moisture sensor based on NaCl/BaTiO_3_ composite films fabricated using a facile aerosol deposition (AD) process. The proposed sensing model operates based on an enormous NaCl ionization effect in addition to natural moisture polarization, whereas all previous sensors are based only on the latter. We obtained an optimal sensing performance in a 0.5 µm-thick layer containing NaCl-37.5 wt% by manipulating the sensing layer thickness and weight fraction of NaCl. The NaCl/BaTiO_3_ sensing layer exhibits outstanding sensitivity over a wide humidity range and a fast response/recovery time of 2/2 s; these results were obtained by performing the one-step AD process at room temperature without using any auxiliary methods. Further, we present a human respiration monitoring system using a sensing device that provides favorable and stable electrical signals under diverse respiratory scenarios.

## 1. Introduction

Recently, progress in healthcare technology for monitoring human respiration has received considerable research attention because of its multifunctional applications in estimating the basic health status of patients [1,2,3]. The prediction of real-time respiration patterns is a vital criterion for characterizing various illnesses, such as pneumonia, asthma, bronchitis, and cardiovascular diagnoses, because these patients have an unstable respiration cycle caused by abnormalities in the respiration rate [4,5,6,7]. The respiration cycle of a healthy adult is 12–20 times per minute during exhalation and inhalation, whereas that of a patient suffering from a fatal disease was reported to be over 24 times per minute (in general hospital wards) with scientific exactitude [8,9]. This implies that it is very important to observe continuous patterns at diverse depths and rates of respiration with accurate control.

In most clinical settings, such as in an emergency room, the conventional method for measuring the breath rate is to detect chest constriction when a patient takes a breath [10]. Although this method is considered very simple because it only involves observing the rising and falling of the chest, it remains challenging to obtain objective and deliberate information for individual patients in diverse scenarios. Several techniques have been developed to observe respiratory activity using multifarious sensing devices. For example, thermal sensors, which include thermistors, detect the temperature difference in human airflow between inhalation and exhalation and transduce the respiratory signal [11]. Further, the respiration signal can be measured using a method in which a pressure sensor attached to the nares with a nasal cannula mask detects pressure fluctuations induced by inhalation and exhalation [12,13]. The major drawback of these methods is that they not only require expensive and cumbersome auxiliary equipment but also are prone to inducing signal errors, as they need to be rigidly fixed in the nares and are often subject to unintentional displacement. An airborne ultrasound system has been suggested to observe the rate of respiration based on the velocity of sound induced by airflow; however, such transduction makes it difficult to perceive sensing information in a variety of scenarios (e.g., coughing, sleeping, and sneezing) and noisy environments [14,15,16].

Alternatively, moisture sensors have been investigated as a means to monitor respiration signals using simple and precise methods [17,18,19]. The exhaled airflow from the human nose or mouth always involves a high humidity of over 90 RH%; the inhaled air induces a dry state with a relatively low humidity range. For sensing humidity, the device generates different electrical signals with high and low humidity levels in terms of capacitance, resistance, and conductance in accordance with the Grotthuss mechanism [20]. Based on this theory, protons (H^+^) dissociated from water molecules (H_2_O) migrate from one vapor molecule to another via strong hydrogen bonding, which is also known as proton hopping. In the high-humidity range required for human exhalation, a sufficient number of self-ionized protons activate charge transport between neighboring water molecules, which results in an increase in the electrical conductivity of the inner sensing structure. However, low humidity (e.g., during inhalation) interferes with charge carrier migration between adjacent sites of water molecules because of the discontinuous physisorbed H_2_O layer; this makes it impossible to generate substantial electrical conductivity. Therefore, the sensitivity deviation caused by two conductive scenarios can easily generate respiratory patterns, which implies that the rate and depth of respiration patterns are dominated by humidity sensitivity and response/recovery time [21,22,23,24].

Moisture sensors for respiration systems have been developed to realize beneficial sensing abilities. For example, nano-sized devices with high porosity have been used to achieve higher sensitivity and rapid response/recovery times based on ceramics and carbon nanomaterials [25,26,27,28]. Further, polymer moisture sensors are advantageous for the easy control of the device structure using a solution process and batch preparation [29,30]. Recently, two-dimensional (2D) materials such as graphene, reduced graphene oxide, and molybdenum disulfide have been widely utilized as humidity-sensing materials [22,24,31,32]. These materials normally facilitate the fabrication of a unique porous structure, which is advantageous for realizing outstanding sensing properties with large surface reaction sites for humidity vapors.

However, most moisture sensors fabricated using a single material (ceramic, polymer, etc.) are limited to realizing visible respiratory patterns in response to both exhalation and inhalation, which is attributed to low humidity sensitivity, slow response/recovery time, or poor stability in high-humidity environments [33,34]. Thus, they can easily cause unstable electrical signals during continuous measurements for a long time, which can result in an inaccurate analysis in diverse respiratory scenarios. Further, although 2D materials are significantly beneficial for sensing, they suffer from poor film-forming ability [35]. Another major issue is the complex processability of the sensing devices. Although these devices are manufactured by wet-spinning, vacuum filtration, and chemical vapor deposition, it is essential to use material synthesis processes and many auxiliary fabrication methods to improve the sensing properties and film-to-substrate adhesion [31,32].

A promising approach to realizing high sensitivity and fast response to moisture can be achieved by mimicking a natural system. Ionic compounds such as sodium chloride (NaCl), which are non-toxic and low-cost, have a strong reactivity with water molecules, and they dissociate into Na^+^ and Cl^−^ ions because of their natural properties of ionic bonding [36]. For a respiration monitoring system, these ions can instantly improve charge transportation in humid environments and, by extension, the interfacial polarization inside the sensing structure. The humidity sensitivity of traditional sensors depends only on the proton hopping mechanism, whereas the NaCl-based moisture sensor can be operated by a double-charge transportation system induced by H_2_O proton hopping and effective NaCl ionization. However, it is difficult to fabricate a single NaCl layer because of its low stability under long-term humid conditions and its poor film-forming ability. Thus, a composite structure combined with NaCl and ferroelectric material is considered a good candidate for a capacitive-type moisture sensor that can effectively utilize the ionization effect. Respiratory signals were identified by capacitance variations under high and low humidity levels.

We report an effective, simple, and low-cost NaCl/BaTiO_3_ moisture sensor for monitoring respiration. The sensor is fabricated by a one-step aerosol deposition (AD) process, which has numerous advantages, such as fast coating speed, film deposition at low vacuum, full-room-temperature process without any heat treatment, and rigid film-to-substrate adhesion [37,38,39]. The AD process has considerable advantages over conventional technologies because a composite film can be easily prepared by simply mixing and loading more than two materials [40,41,42]. The AD principle is defined by the hammering effect, wherein high-speed particles continuously collide with pre-deposited layers; consequently, the internal density is dominated by the layer thickness and mechanical properties of each material [43,44]. In this work, we designed a unique NaCl/BaTiO_3_ moisture sensor for respiration monitoring by exploring the effect of layer thickness on the sensing properties and by optimizing the weight fraction of NaCl. After characterizing the humidity sensing capabilities, a respiration monitoring test was performed on subjects wearing a portable medical oxygen mask. Considering the facile production method and precise control under diverse respiratory conditions, the proposed moisture sensor holds great potential as a promising approach for realizing novel medical monitoring devices.

## 2. Materials and Methods

### 2.1. Fabrication of NaCl/BaTiO_3_ Composite Films via AD Process

BaTiO_3_ powder (SBT-045B, Samsung Fine Chemical, Ulsan, South Korea) with an average particle diameter of 0.5 µm was used as the starting composite powder in conjunction with commercial NaCl powder (Daejung Chemical & Metals Co., Ltd., Gyeonggi-Do, Korea) with a 3.0 µm diameter. The two powders were weighed in accordance with the proportions of NaCl-25, 37.5, 50, and 75 wt% (i.e., NaCl:BaTiO_3_ = 25:75, 37.5:62.5, 50:50, and 75:25) using a precision balance. Further, the BaTiO_3_ single layer was deposited to compare its elemental distribution with that of the NaCl/BaTiO_3_ composite film, whereas a single NaCl layer was not employed because of its meaningless capacitance value and poor film-forming ability.

The AD apparatus comprises vacuum pumps (rotary pump and mechanical booster pump), a deposition chamber, an aerosol chamber, a moving X-Y stage, and a mass flow controller (MFC). After percolating the prepared composite powders with a fine sieve net, 100 g of the powders was placed in the aerosol chamber. They were vigorously aerosolized by controlling the MFC controller with nitrogen gas at 8 L/min (purity: 99.99%). The blown aerosol was transferred through a Teflon tube to a stainless-steel nozzle at high speed. Then, the aerosol was accelerated by pressure diffusion into the deposition chamber, which was evacuated by vacuum pumps in advance. The accelerated particles impinged on the interdigital electrode (IDE) substrate were located 5 mm away from the nozzle, and they formed a composite film with strong film-to-substrate and particle-to-particle cohesions. Consequently, we successfully obtained AD-prepared NaCl/BaTiO_3_ composite films in a short time of approximately 2 min per sample. All samples were fabricated under the same experimental conditions. The geometric parameters of the IDE substrate were confirmed in our previous research [45].

### 2.2. Characterizations

A humidity-sensing setup was used to observe humidity-sensing properties and to monitor respiration. The setup comprised a humidity chamber (TH-ME-025, Jeiotech Co., Ltd., Seoul, Korea), impedance analyzer (4192A, Agilent Co., Ltd., Santa Clara, CA, USA), micro vacuum probe station, and computer. The capacitance was continuously measured in real time using an impedance analyzer when the humidity changed from 35 RH% to 85 RH%, and vice versa. The humidity sensitivity (S) was calculated as S = ΔC/ΔRH, where ΔC denotes the capacitance deviation at 85% and 35% RH. The response/recovery time was measured under a drastic humidity change from/to atmospheric humidity to/from relatively high humidity (75 RH%) while recording the capacitance per second. Real-time respiration monitoring was conducted with the consent of all subjects. X-ray diffraction (XRD, X’Pert PRO diffractometer, PANalytical, New York, NY, USA) was performed to confirm the crystallinity and crystallite size of the composite films using Cu Kα radiation (~1.54056 Å) over a 2θ range of 20–80°. Surface microstructures of the NaCl/BaTiO_3_ composite films were observed using field-emission scanning electron microscopy (FE-SEM; S-470, Hitachi Ltd., Tokyo, Japan) at 5 kV. Energy-dispersive spectroscopy (EDS) was used to confirm the elemental distribution on the NaCl/BaTiO_3_ composite film surface.

## 3. Results and Discussion

### 3.1. Sensing Properties via Layer Thickness Control

A NaCl/BaTiO_3_ composite film with a NaCl content of 50 wt% (NaCl-50 wt% film) was fabricated on the IDE substrate using the AD process to confirm the deposition feasibility of the proposed sensing layer. The deposition range was 4 mm × 9 mm, which is the actual sensing area for moisture. As shown in Figure 1a, the chemical composition of each element was uniformly distributed over the surface; it forms organically tight connections between particles via the continuous impingement of high-velocity aerosols on the pre-deposited layers.

Figure 1b depicts the comparative XRD patterns of NaCl-50 wt% and BaTiO_3_ single films employed on a platinum (Pt) substrate, which exhibits intrinsic phases (BaTiO_3_ and NaCl) with no transformation and a peak shift attributed to the room-temperature fabrication method. Further, we analyzed the crystallite size of the two types of films because NaCl has different mechanical properties from those of BaTiO_3_ ceramic. The crystallite size was calculated from the full width at half maximum (FWHM) values at a peak position (2θ) of ~45°. The size of the BaTiO_3_ single film was ~7.83 nm, whereas that of NaCl-50 wt% film was ~9.89 nm and ~294 nm for BaTiO_3_ and NaCl, respectively. The major difference between the BaTiO_3_ and NaCl crystallite sizes can be inferred from the intrinsic mechanical properties of each material, i.e., Mohs hardness and density. Effective particle pulverization in the AD process is predominantly influenced by hardness and density because film formation and resultant densification can occur because of the severe hammering effect of high kinetic-energy particles [40,46,47]. Thus, the loading powder in the AD process is preferred for ceramic materials. Compared with BaTiO_3_ particles, NaCl has a lower hardness and lower density [48,49]. The pulverization of BaTiO_3_ particles is attributed to their high bombardment energy with the substrate, whereas NaCl particles cannot be pulverized well because of the low collision force. Thus, the reduction in the size of BaTiO_3_ and NaCl is approximately 50 times and 10 times, respectively, in comparison with the initial particle size.

Inspired by the successful deposition of NaCl/BaTiO_3_ composite films, we attempted to employ NaCl/BaTiO_3_ composite films with different layer thicknesses (0.5, 1.0, and 1.5 µm) to find the optimal thickness suitable for sensing capabilities. Figure 2 shows the humidity sensitivities and response/recovery times, which are considered to be crucial factors for respiration monitoring, for the three types of samples. Humidity sensitivity was repeatedly measured to define the stability; however, it showed no degradation performance or NaCl dissolution after the test because NaCl is not only physically stable in the gas phase of H_2_O but also protected by the BaTiO_3_ matrix [50]. In addition, the humidity sensitivities are the mean values of repeatability tests, and the measured data had minor error rates. Remarkably, NaCl/BaTiO_3_ sensing layers showed a high sensitivity of over 250 pF/RH% despite the one-step room-temperature process, whereas the aerosol-deposited BaTiO_3_ single layer demonstrated only 1.39 pF/RH% [51]. The previous BaTiO_3_ moisture sensor essentially required an auxiliary method such as post-annealing to improve the humidity sensitivity; however, NaCl-based ionization has a considerably more efficacious potential for charge separation than any other treatment. Thus, massive interfacial polarization is generated in the inner structure, which leads to a drastic increase in sensitivity under humid conditions.

The 1.0 µm-thick film had the highest sensitivity of 1413 pF/RH% among the three samples, whereas the others were measured with similar sensitivity values. The cause of the discrepancy in these sensitivities can be ascribed to the amount of NaCl and the AD densification mechanism. The densification of the surface and internal microstructure rapidly progressed with an increase in thickness because NaCl/BaTiO_3_ particles continuously accumulated on the pre-deposited layer. However, a humidity sensor requires a porous structure to allow moisture to effectively permeate into the inner film, which means that high densification due to increased thickness has a negative effect [52,53,54,55]. This indicates that the amount of NaCl embedded in the composite film proportionally increases with layer thickness, and this can lead to a higher sensitivity due to substantial NaCl ionization-based interfacial polarization. Thus, there is a trade-off relationship between layer densification (negative effect) and charge transportation (positive effect) with increasing film thickness with respect to humidity sensitivity. The high sensitivity of the 1.0 µm-thick film was predominantly controlled by the ionization effect, but it was less influenced by the layer densification up to 1.0 µm thickness. Although a 1.5 µm-thick film has plenty of NaCl molecules that can induce charge separation, further densification blocks the open pores on the surface for moisture absorption. As shown in Figure 3a,c, the 0.5 µm-thick film contains large amounts of nanopores over the surface, whereas the surface microstructure of the 1.5 µm-thick film displays strong bonding between particles, which forms few pores.

A short response/recovery time was achieved in the lower thickness range despite the high sensitivity of the 1.0 µm-thick film. The response time was measured under a sudden change in humidity: from ambient humidity (~25 RH%) to 75 RH%. Then, the response time was calculated when the capacitance at 75% RH was saturated to a similar value with an error rate below 0.1% for several seconds. The 0.5 µm-thick film showed the shortest response/recovery time (3/5 s) among the three samples, whereas composite layers with thicknesses greater than 1.0 µm had a slow time in both cases. In the aerosol-deposited NaCl-based sensing layers, the response/recovery time was affected by the structural surface and inner density. The 0.5 µm-thick film is prone to effectively absorbing and desorbing moisture despite drastic changes in humidity because it involves a large quantity of open pores and a large pore volume on the outside and inside of the structure [56,57]. Accordingly, the absorbed moisture can rapidly react with NaCl particles and thus generate immediate electrical conductivity, whereas the desorbed moisture can be effectively evaporated through the capillary structure. Consequently, the final capacitance quickly stabilizes even if the humidity is abruptly changed. We verified that the response/recovery time of the 1.0 µm-thick film is 4 s/14 s. The recovery speed steadily decreased until the test was completed, and it showed no steep signal changes during the measurement. This proves that films with over 1.0 µm thickness cannot completely desorb moisture from the inner structure because of the relatively high densification; however, NaCl rapidly reacts with moisture. The 1.5 µm-thick film is formed with surface flatness and a resultant high internal density, as shown in Figure 3c; therefore, the reduction ratio of both the response and recovery times becomes too slow to stabilize the capacitance at ambient humidity.

Further, the response time is shorter than the overall recovery time. This result demonstrates that in the case of H_2_O adsorption, moisture can instantly react with NaCl, which forms Na^+^ and Cl^−^ ions, and then it can activate interfacial polarization by charge transportation. However, the recombination of Na^+^ and Cl^−^ ions is considerably slower than the NaCl ionic reaction with H_2_O when moisture is desorbed from the sensing layer. Proton hopping based on the Grotthuss mechanism is ineffective for rapidly decreasing the chemical potential energy to achieve an equilibrium state from the disequilibrium state.

The normal respiration of a volunteer was monitored using a portable medical oxygen mask to confirm the influence of each layer thickness on the respiration signals with the three types of sensing films (NaCl 50 wt%) in the range of 0.5–1.5 µm thickness. The sensing device was placed in the ventilation holes of a medical oxygen mask. Then, a volunteer put on the mask and breathed at a constant speed, simultaneously recording real-time capacitance signals. The respiration cycle includes the same periods of exhalation and inhalation. First, respiration monitoring with a period of 8 s was performed to confirm the feasibility of the signal variations. As shown in Figure 4a–c, the 0.5 µm-thick film detected the features of respiration, whereas the 1.0 and 1.5 µm-thick films could not identify fine fluctuations from either exhalation or inhalation.

This discrepancy in signal variations can be inferred from the response/recovery time rather than humidity sensitivity. The 0.5 µm-thick film exhibited a relatively fast response/recovery time, and it could quickly react to changes in the amount of moisture. Although the 1.0 µm-thick film had a similar response time to that of the 0.5 µm-thick film, it is not effective for an immediate signal drop in inhaled air because of the slow recovery time of 14 s. Further, the response/recovery speed is significantly degraded in the case of the 1.5 µm-thick film in Figure 2, which makes it difficult to achieve a capacitance variation in 4 s/4 s of exhaled and inhaled air, which indicates poor respiratory patterns. All sensors exhibited consistent capacitance variations during the test when a longer respiratory cycle of 16 s was applied to the three sensing devices. Since the 1.0 µm-thick film had the highest humidity sensitivity among our samples, its normalized capacitance variation showed the highest value, even though the recovery time was slow. Although the 0.5 and 1.5 µm-thick films had an almost identical humidity sensitivity, the former had a higher capacitance dynamic range, which is attributed to the large gap in their response/recovery time. Further, the capacitance decreased slowly from the inhaled air compared to the increasing rate in exhalation, which is confirmed by the recovery tendency of the films with over 1.0 µm thickness.

The effect of aerosol-deposited layer thickness on sensing capabilities and breathing detection clearly confirmed that the 0.5 µm-thick sensing layer could effectively detect respiration signals in diverse breathing situations because of its favorable humidity sensitivity and response/recovery time due to the NaCl ionization effect and its nanoporous structure.

### 3.2. Optimization of NaCl Ratio for Real-Time Monitoring

Although the 0.5 µm-thick NaCl/BaTiO_3_ composite film could realize better respiratory signals compared to other samples with over a 1.0 µm-thickness, it still has a considerable limitation in terms of observing the signal of abnormal breath because the response/recovery time of 3/5 s makes it difficult to cover fast respiration cycles shorter than 4 s. Further, it is essential to monitor a rapid breath rate for up to 2 s to detect stable respiratory patterns for most patients with a fatal illness. We attempted to fabricate four types of 0.5 µm-thick NaCl/BaTiO_3_ composite films by changing the content of NaCl from 25 wt% to 75 wt% with a constant gap, considering that (1) the recovery speed of Na^+^ and Cl^−^ is considerably delayed until they reach the equilibrium state (NaCl), as shown in Figure 2, and (2) structural porosity is influenced by NaCl in the AD mechanism. Then, we evaluated the sensing capabilities.

Figure 5 depicts the comparative sensing capabilities involving the humidity sensitivity and response/recovery time for sensors with different NaCl ratios. The humidity sensitivity was significantly improved by increasing the NaCl ratio in the composite film. The sensitivity of sensing layers containing less than NaCl-50 wt% was enhanced at a constant rate of approximately 4 times, whereas the NaCl-75 wt% film showed an extremely high sensitivity of 4535 pF/RH% compared to the NaCl-50 wt% film (approximately 18 times). Simultaneously, both the response and recovery times decayed with an increasing NaCl ratio, which implies that the NaCl filler in the composite film resulted in a trade-off relationship between the sensitivity and humidity reaction time.

The unique structural variation controlled by NaCl played a dominant role in promoting the humidity sensitivity of the AD-prepared NaCl/BaTiO_3_ composite films. Figure 6 shows that even though NaCl-0, 25, 50, and 75 wt% films with 0.5 µm thickness were fabricated under the same experimental conditions, they were classified into different growth mechanisms as a transitional density structure with an increase in the NaCl ratio.

The surface morphology of the BaTiO_3_ single layer shows high densification with no pores and a flattened shape (Figure 6a), whereas the NaCl-25 and 50 wt% films presented a relatively rough surface forming plenty of non-fractured particles (Figure 6b,c). We observed a large quantity of both pores and agglomerated sizable NaCl particles on the surface when NaCl-75 wt% NaCl was contained in the composite films. The surface microstructure in the AD process can be elucidated based on the hammering effect, which generates a dense structure via the consecutive impact of high-kinetic-energy particles onto the pre-deposited films [58,59,60]. Thus, a reasonable hammering effect is ascribed to the mechanical hardness and intrinsic density of the starting particles [58,61]. NaCl, with low density and low hardness, is not sufficient to affect the effective pulverization of particles despite its brittleness. However, ceramics such as BaTiO_3_ with high mechanical hardness and suitable density can severely impinge on the substrate and the pre-deposited layers, which can contribute to the high structural densification. Previous researchers showed the influence of soft PTFE with low density and hardness on the growth of aerosol-deposited ceramic-based composite films [62,63,64,65]. Their results revealed that PTFE was sufficient to interfere with the strong impaction and particle pulverization, and it consequently formed a large quantity of pores and debris, as well as a rough surface. Fortunately, in the field of moisture sensors, these structural properties induced by insufficient hammering effects are beneficial for increasing humidity sensitivity because they provide moisture reaction sites through numerous open pores and large pore sizes on the surface and internal film, respectively. Thus, a high NaCl content can elevate the humidity sensitivity in our composite films. However, NaCl contents in the range of 0–50 wt% cannot be considered the dominant cause of the degraded quality of the layer structure, which means that NaCl ionization is a key factor for increasing the sensitivity rather than structural variation in this NaCl range. This is because there are no large differences in surface structure among the three samples, as shown in Figure 6a–c, except for non-fractured particles. Accordingly, humidity sensitivity steadily increased with the NaCl ratio. In this range of the NaCl content, the response/recovery time steadily decays from 1/2 s to 3/5 s in Figure 5a–c, which indicates that dissociated Na^+^ and Cl^−^ ions in a highly humid environment require considerable elapsed time to return to the equilibrium state of NaCl.

In the case of the NaCl-75 wt% film, the sensing capabilities were strongly affected by a dual effect, i.e., NaCl ionization and porous structure. Compared to other films with NaCl-50 wt%, the NaCl-75 wt% film formed a unique surface morphology with a vivid micro-porosity as if it were deposited using different materials or fabrication processes. The surface was composed of a large amount of sizable NaCl (2.0–3.0 µm in Figure 6d), which was almost similar to the initial particle size. This growth mechanism can be interpreted as excessive levels of NaCl in the composite film, which considerably hinder the effective pulverization of both NaCl and BaTiO_3_. Although film densification is controlled by the hammering effect, it is significantly influenced by particle-to-particle collisions [66,67]. However, it is difficult for NaCl particles to receive high kinetic energy from BaTiO_3_ during the AD process because the NaCl-75 wt% film contains a small amount of BaTiO_3_ with high hardness. In addition, the pre-deposited layer is composed of NaCl, and therefore, NaCl-to-NaCl collisions predominate over BaTiO_3_-to-NaCl collisions, which give rise to a porous structure. The humidity sensitivity increased significantly compared to that of the NaCl-50 wt% film, whereas the response/recovery speed became very slow because of the large amount of NaCl.

Respiratory patterns were observed by changing the exhalation/inhalation cycle with the four types of NaCl/BaTiO_3_ composite films containing different NaCl ratios. Considering that the respiratory rate and depth are critical factors for evaluating the physiological and physical conditions of patients with breathing diseases, various respiration states were monitored for 2 to 12 s. As shown in Figure 7a, NaCl-25 and 37.5 wt% films detected stable respiratory signals even at an exhalation/inhalation rate of 1/1 s.

Since the NaCl-37.5 wt% film has a higher sensitivity than the NaCl-25 wt% film, its signal variation is more distinct despite the slightly slow response/recovery time. These variation rates are in line with the excellent data recently reported on moisture sensors focused on respiratory patterns (Table 1) [33,68,69,70,71,72,73]. In addition, although the humidity sensitivities of the NaCl-50 and 75 wt% films are significantly high, fast respiratory patterns cannot be observed due to their slow response/recovery times. The respiratory patterns of the NaCl-50 wt% film are observed when the respiratory cycle is increased up to 12 s (Figure 7b,c); the NaCl-75 wt% film, with the highest sensitivity among our samples, does not show a clear signal variation despite a longer breathing period of 12 s. In addition, although the NaCl-25 wt% film had a very fast response/recovery time of 1/2 s, respiration monitoring was not observed in the case of a longer breathing period because of its low sensitivity, which easily causes unstable signals under sudden humidity changes. This result demonstrates that humidity sensitivity is important for realizing real-time respiration monitoring; however, the electrical conductivity under a sudden humidity change should have a quick response/recovery time.

Consequently, the aerosol-deposited NaCl/BaTiO_3_ moisture sensor with a NaCl content of 37.5 wt% and 0.5 µm thickness showed outstanding feasibility for real-time respiration monitoring, highlighting its sensing capabilities and ability to monitor diverse breathing states.

## 4. Conclusions

We present an ionically polarizable moisture sensor that comprises a NaCl/BaTiO_3_ composite film fabricated solely by the aerosol deposition process; this leads to outstanding sensing performance and the stable detection of respiratory signals. We confirmed that NaCl-induced interfacial polarization is beneficial for increasing humidity sensitivity by comparing the NaCl/BaTiO_3_ composite film and BaTiO_3_ single film. From the effect of layer thickness on sensing capabilities, a 0.5 µm-thick film exhibited both high sensitivity and fast response/recovery time because of the formation of a nanoporous structure by a unique AD growth mechanism, which resulted in an immediate moisture reaction to human breath. We demonstrated the correlation between the weight fraction of NaCl in the composite film and sensing capabilities by optimizing the NaCl ratio of the device. Although the high content of NaCl led to a drastic rise in humidity sensitivity because of the increase in the ionization effect, the response/recovery time became very slow and unsuitable for detecting abnormal breathing patterns. The 0.5 µm-thick film containing NaCl-37.5 wt% even reacted to a quick exhalation/inhalation of 1 s, whereas others were not favorable for observing fast or slow respiratory cycles because of their insufficient response/recovery time and sensitivity.

We believe that this research may provide a more suitable method for operating ionization-based moisture sensors with high sensitivity and fast response compared to previous moisture sensors.

## Figures and Tables

**Figure 1 sensors-22-05178-f001:**
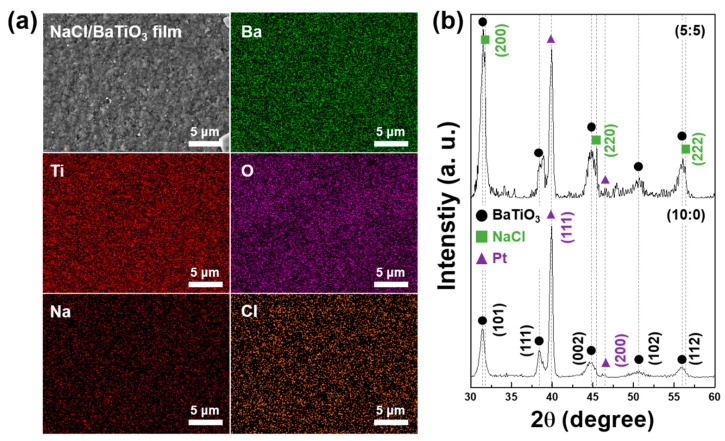
(**a**) SEM micrograph and EDS images of NaCl/BaTiO_3_ (50:50) composite film. (**b**) XRD analysis of as-deposited BaTiO_3_ single layer and NaCl/BaTiO_3_ (50:50) composite film.

**Figure 2 sensors-22-05178-f002:**
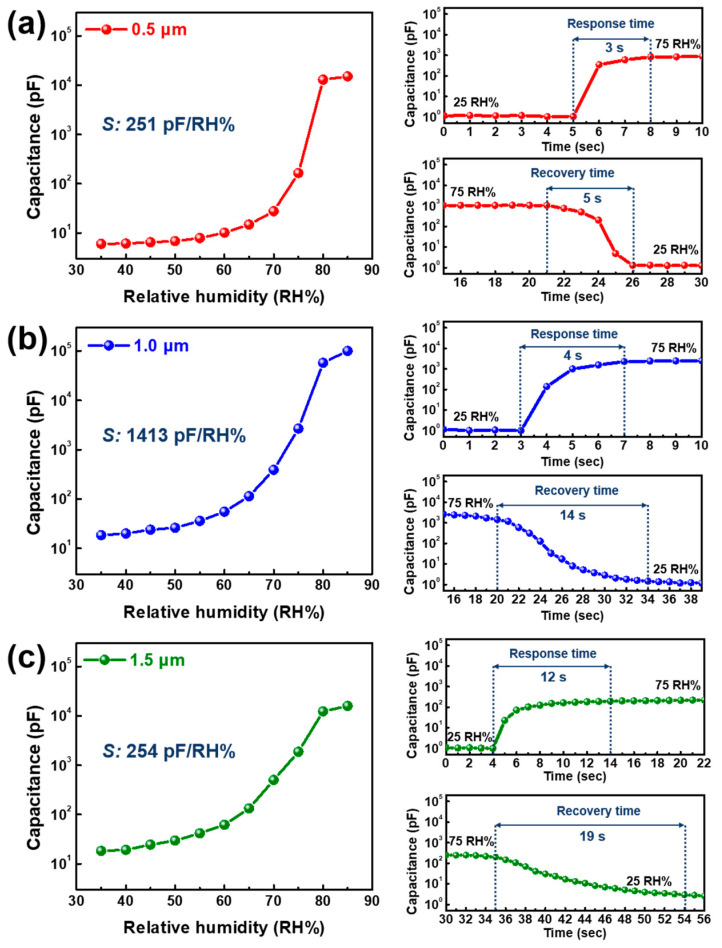
Humidity sensing abilities with different RH values ranging from 35 RH% to 85 RH% for (**a**) 0.5, (**b**) 1.0, and (**c**) 1.5 µm-thick composite layers. Response and recovery times for NaCl/BaTiO_3_ composite films were measured under a sudden humidity change from ambient humidity to 75 RH% and vice versa.

**Figure 3 sensors-22-05178-f003:**
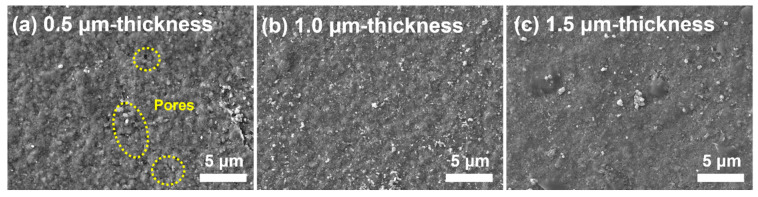
Variation in the surface microstructures of NaCl/BaTiO_3_ composite film when gradually increasing layer thickness to (**a**) 0.5 µm, (**b**) 1.0 µm, and (**c**) 1.5 µm.

**Figure 4 sensors-22-05178-f004:**
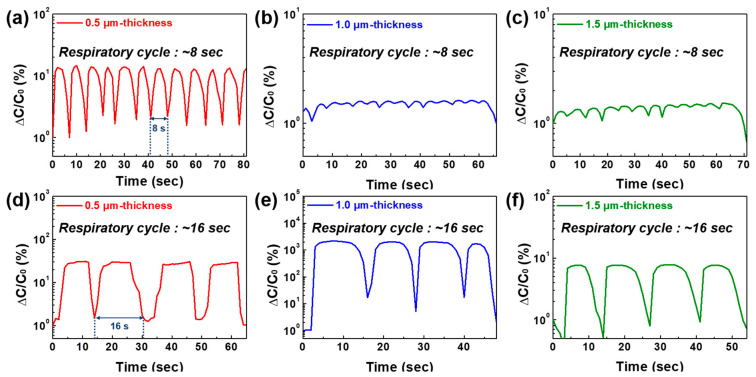
Respiration monitoring of NaCl/BaTiO_3_ humidity sensors with different layer thicknesses of 0.5, 1.0, and 1.5 µm. Each humidity sensor monitored two types of respiratory cycles of (**a**–**c**) 8 s and (**d**–**f**) 16 s.

**Figure 5 sensors-22-05178-f005:**
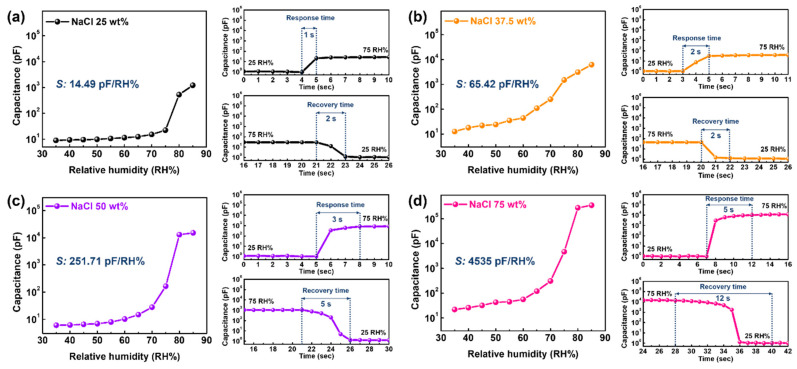
Comparative results of humidity sensitivity and response/recovery time for NaCl/BaTiO_3_ composite films containing different NaCl ratios of (**a**) 25, (**b**) 37.5, (**c**) 50, and (**d**) 75 wt%, respectively.

**Figure 6 sensors-22-05178-f006:**
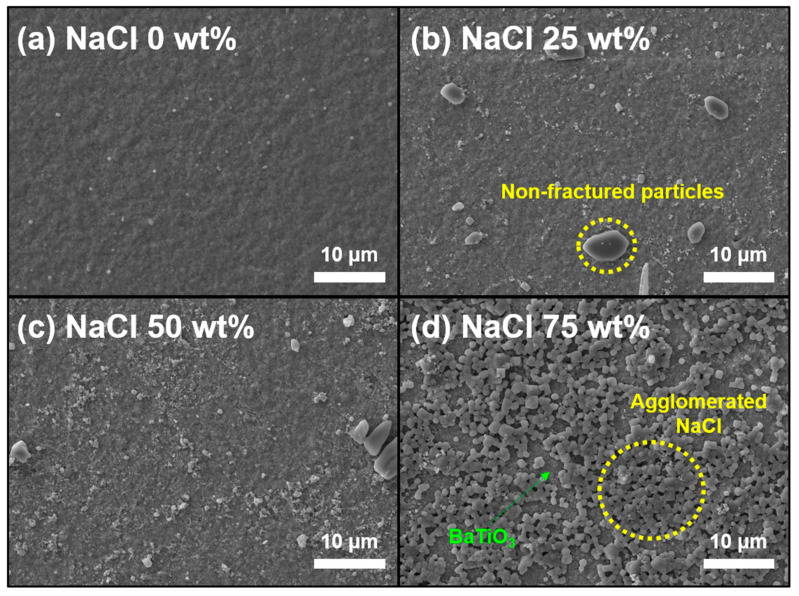
SEM images showing the surface microstructures of NaCl/BaTiO_3_ composite films with (**a**) 0, (**b**) 25, (**c**) 50, and (**d**) 75 wt% NaCl, respectively.

**Figure 7 sensors-22-05178-f007:**
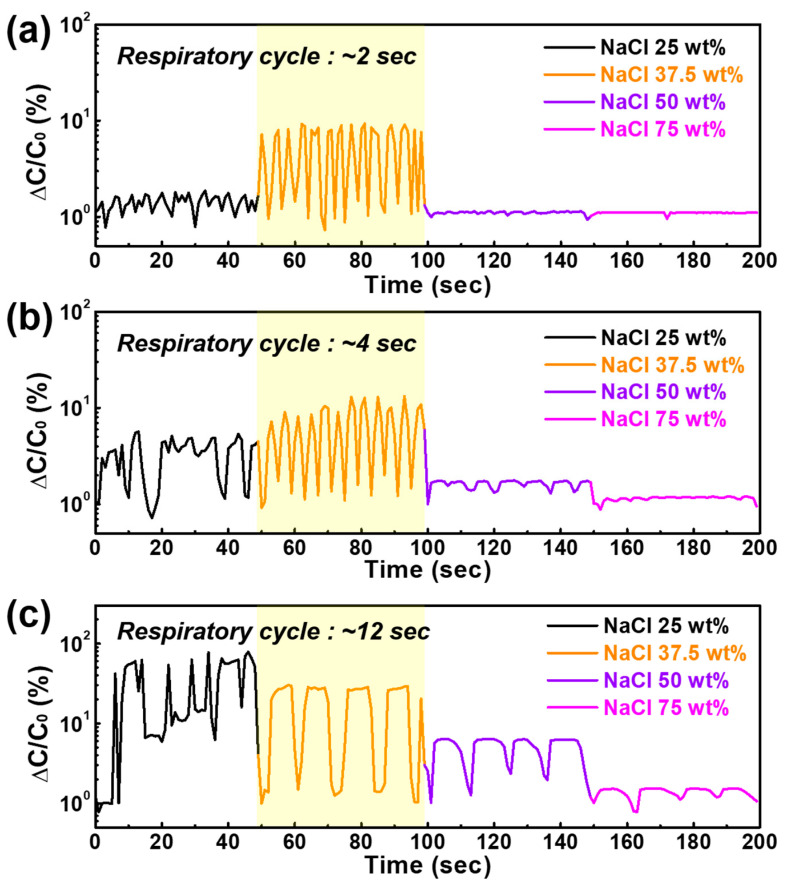
Normalized capacitance variations of the four types of NaCl/BaTiO_3_ composite films with 0, 25, 50, and 75 wt% NaCl in response to various respiration cycles of (**a**) 2 s, (**b**) 4 s, and (**c**) 12 s.

**Table 1 sensors-22-05178-t001:** Comparing previously reported signal variation rates and fabrication processes.

No.	Material	Process	Type of Signal	Signal Variation Rate [%]
1	GO [33]	Chemical vapor deposition	Resistance	3
2	PEDOT-PSS [68]	Spin coating	Resistance	6
3	PMDA-ODA PAA [69]	Laser writing	Resistance	1
4	Graphite [70]	Solution	Current	2
5	Cellulose and CNT [71]	Solution	Current	65
6	Si nanocrystals [72]	Co-sputtering	Current	2.2
7	Cellulose [73]	Pencil drawing	Capacitance	30
8	NaCl/BaTiO_3_(This work)	AD process	Capacitance	54

## Data Availability

The data presented in this study are available on request from the corresponding author.

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
