# Peer review of "NaCl Ionization-Based Moisture Sensor Prepared by Aerosol Deposition for Monitoring Respiratory Patterns"

_sensors, 2022, doi:10.3390/s22145178_

Round 1
Reviewer 1 Report
The authors present a novel type of humidity sensor based on NaCl-ionization-based with a fast time response. This type of sensor might be of interest for the stated application of human respiration monitoring as well as several other applications requiring fast time response. A detailed discussion of the results is missing however as well as some critical details in the materials and method section (e.g. nr of replicates).
-p7 'film contains large amounts of nanopores over the surface'. This might not be clear for all readers from the photos shown in Fig 3.
P9 'Their results revealed that PTFE was sufficient to interfere with the strong impaction and particle pulverization' Not clear what is meant here.
-How has the response time been defined in this study (e.g. T90)?
-Comment on the (potential) long term stability of the developed sensor.
-For the sensor sensitivity data of the sensors under test, what is missing are the standard deviations (how many repeats?) and the sensor noise in order to calculate the resolution of the sensors. Throughout the paper details are missing on number of replicates etc.
-No comparison is made between the outcome of this study and previous results for similar sensors in literature. In general, the conclusion section is
- p1 'The proposed sensing model is operated by an enormous NaCl ' Not clear what is meant here with operated.
-Fig 2: How long does it take for the humidity to change? Has this been checked by a reference method?
-Check the number of significant digits in throughout the document (e.g. 1,413 pF/RH% on p6)
-In Fig 4 what is the wt% of NaCl?
Author Response
Dear Reviewer,
First of all,
I would like to thank you very much for your sincere comments and hope to meet the comments.
I also think it was a great opportunity to improve this article.
I wrote each answer of the comments you had pointed out in this file (Response to Reviewers).
And, in revised manuscript, I inserted the corresponding content and figures marked in red.
I am looking forward to your reply.
Thank you again.
Yours sincerely,

Reviewer 2 Report
1. The work is very relevant and truly novel. But some more information would really enhance the quality of the work.
2. No information on the pore size and the crystallite size is there. Also It would be really logical to correlate the sensing properties with the structural properties (pore size , crystallite size etc) so as to understand the sensing mechanism in a better way.
3. In the fabrication section, a schematic of the experimental design is missing. If possible, give reference to earlier articles published by the group.
4. The 37.5Wt % Nacl is found to perform well, but no corresponding SEM data has been provided. please specify?
Author Response

(The authors gave the same response as above.)
